# Covariations between pupil diameter and supplementary eye field activity suggest a role in cognitive effort implementation

**Julien Claron[1,2], Julie Royo[2], Fabrice Arcizet[3], Thomas Deffieux[1], Mickael Tanter[1], Pierre Pouget[2]* **

**1** Physics for Medicine, ESPCI, INSERM, CNRS, PSL Research University, Paris, France, **2** Institut du Cerveau, INSERM 1127, CNRS 7225 Sorbonne Université, Paris, France, **3** INSERM, CNRS, Institut de la Vision, Sorbonne Université, Paris, France

☯ These authors contributed equally to this work.
* pierre.pouget@upmc.fr, pierre.pouget@icm-institute.org

**Data Availability Statement:** All data files are available from the database: https://osf.io/2q357/?view_only=7380834a5bc84df38a569410eecbbabb.

## Abstract

In both human and nonhuman primates (NHP), the medial prefrontal region, defined as the supplementary eye field (SEF), can indirectly influence behavior selection through modulation of the primary selection process in the oculomotor structures. To perform this oculomotor control, SEF integrates multiple cognitive signals such as attention, memory, reward, and error. As changes in pupil responses can assess these cognitive efforts, a better understanding of the precise dynamics by which pupil diameter and medial prefrontal cortex activity interact requires thorough investigations before, during, and after changes in pupil diameter. We tested whether SEF activity is related to pupil dynamics during a mixed pro/antisaccade oculomotor task in 2 macaque monkeys. We used functional ultrasound (fUS) imaging to examine temporal changes in brain activity at the 0.1-s time scale and 0.1-mm spatial resolution concerning behavioral performance and pupil dynamics. By combining the pupil signals and real-time imaging of NHP during cognitive tasks, we were able to infer localized cerebral blood volume (CBV) responses within a restricted part of the dorsomedial prefrontal cortex, referred to as the SEF, an area in which antisaccade preparation activity is also recorded. Inversely, SEF neurovascular activity measured by fUS imaging was found to be a robust predictor of specific variations in pupil diameter over short and long-time scales. Furthermore, we directly manipulated pupil diameter and CBV in the SEF using reward modulations. These results bring a novel understanding of the physiological links between pupil and SEF, but it also raises questions about the role of anterior cingulate cortex (ACC), as CBV variations in the ACC seems to be negligible compared to CBV variations in the SEF.

## Introduction

Seminal studies revealed that pupil dilation varies with increasing task demands, including perception, attention, task consolidation, learning, and memory [1–6]. Two dominant interpretations for these findings have been proposed. Numerous authors concluded that pupil

**Funding:** MT is PI of a research grant from the European Research Council (ERC) under the European Union's Seventh Framework Program (FP7/2007–2013)/ERC Advanced grant agreement no. 339244-FUSIMAGINE. MT is granted by the Fondation Bettencourt-Schueller. MT and TD are also supported by the Inserm Accelerator of Technological Research in Biomedical Ultrasound. The funders had no role in study design, data collection and analysis, decision to publish, or preparation of the manuscript.

**Competing interests:** We have read the journal's policy and the authors of this manuscript have the following competing interests. TD and MT and co-inventors of several patents in the field of functional ultrasound neuroimaging and co-founders of Iconeus company which commercializes ultrasonic neuroimaging scanners.

**Abbreviations:** ACC, anterior cingulate cortex; CBV, cerebral blood volume; FEF, frontal eye field; fUS, functional ultrasound; GLM, generalized linear model; LC, locus coeruleus; PFC, prefrontal cortex; PLR, pupil light reflex; rCBV, relative CBV; ROI, region of interest; SMA, supplementary motor area; SC, superior colliculus; SEF, supplementary eye field.

dilation reflects the demands of a task, whereas others took it a step further and proposed that pupil dilation reflects the effort exerted in response to such demands [1,2,7]. The precise neural substrates by which such cognitive processes influence pupil diameter are still unclear, but inputs from the dorsal part of the medial prefrontal cortex (dmPFC), which mediates arousal, are likely involved.

The dmPFC contains the frontal eye fields (FEF), supplementary motor area (SMA), and supplementary eye field (SEF). The FEF is known to be involved in the control of eye movements and attention, and recent studies have shown that the amplitude of pupil responses depends on the combination of the light stimulus and subthreshold FEF electrical microstimulation [8,9]. Strongly interconnected to the FEF and anterior cingulate cortex (ACC), the SEF is a key region that integrates attentional, short-term memory, and oculomotor tasks [10,11]. The SEF also directly projects to the brainstem oculomotor nucleus. dPMC, including the ACC, SEF, and FEF networks may directly modulate the olivary pretectal nucleus, which encodes retinal illumination and directly activates the pupil-constrictor pathway [8,12–14]. In addition, the ACC, SEF, and FEF networks may act indirectly through the occipital visual cortical areas, or superior colliculus (SC), in which the visual responses are modulated by FEF and may, in addition to programming the oculomotor plan, participate in the pupil light reflex (PLR) [15–18]. Although the function of the SEF in oculomotor tasks is reasonably well defined, involvement of SEF activity in the frontal controller circuit of pupil dynamics is still unknown.

Pupil dynamics have been studied during preparation and before the execution of eye movements during oculomotor protocols [19] providing unique insights into the neuronal substrate coordinating cognitive processing, sensor-motor transformation, and pupil diameter. In the context of the antisaccade task, subjects are instructed before the appearance of a stimulus to either automatically look at the peripheral stimulus (prosaccade) or suppress the automatic response and voluntarily look in the opposite direction from the stimulus (antisaccade). In this type of paradigm, pupil diameter was found to be larger in preparation for correct antisaccades than in preparation for correct prosaccades and erroneous prosaccades made in the antisaccade condition [20]. When an incorrect saccade is executed with latencies in the range of express saccades, execution of the movement indicates that subjects are unable to inhibit involuntary actions, whereas they have no difficulties in generating voluntary saccades if they correct such directional errors. In humans, during saccade preparation, pupil size appeared to be larger in preparation for correct antisaccades than in preparation for correct prosaccades. Given that the SEF is known to be critically involved in the production of antisaccades [21], the precise dynamics through which pupil diameter and SEF activities are conjugated merits further investigation before, during, and after pupil diameter modulation. In a previous study on short sessions (of about 200 seconds), we noticed different slopes and different evolution of CBV depending on the sessions, the subjects and tasks (see Fig 1 from Dizeux and colleagues [22]). In order to understand the origin of these variations, we decided in this new study to study and manipulate specifically these variations of CBVs during a session: in much longer sessions (more than 2,000 seconds), with a randomization of the nature of the trials and by manipulating the level of reward.

This question can be addressed using modern neuroimaging techniques, such as functional ultrasound (fUS) imaging. This innovative imaging technique allows very precise mapping of fine temporal changes in brain neurovascular activity at high spatial resolution in large cortical areas in nonhuman primates [22,23]. In the present series of experiments, we tested whether pupil dynamics are linked to SEF activity during an antisaccade task on awake monkeys.

We obtained 2 primary results: (1) SEF activity is a robust predictor of specific variations of pupil diameter over both short (milliseconds) and long (minutes) time scales; and (2) strong

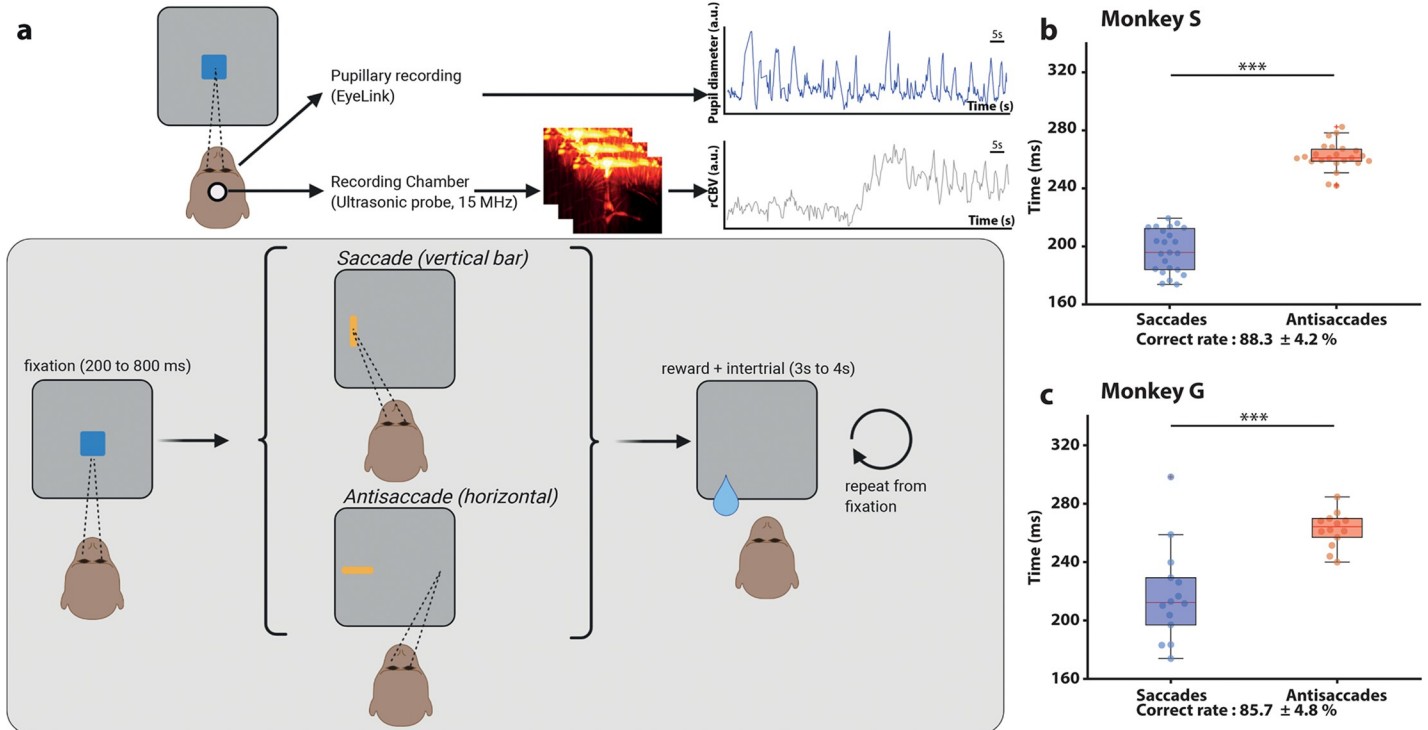

**Fig 1. Task timeline.** (a) The heads of the monkeys are fixed in a chair with a 15-MHz ultrasonic probe in a recording chamber. An EyeLink recording system records the eye position and pupillary diameter in real time. During the task, there is a baseline of 200 to 220 s (randomized) and a fixation point is shown. If the animal succeeds, a prosaccade (vertical rectangle) or an antisaccade (horizontal rectangle) is shown on the screen. Based on the cue, the animal performs a saccade or an antisaccade and, if he succeeds, receives a reward associated with a specific color of the fixation point (red: 0.5 times the normal reward, blue: 1 time the normal reward, green: 1.5 times the normal reward). This action is followed by a gray screen of 3 to 4 s (randomized) used as an intertrial before repeating from the fixation point. (b) The average saccade response time for Monkey S for all sessions was 197.0 ± 15 ms, and the antisaccade response time 262.0 ± 10 ms, with a total correct rate of 88.2 ± 4.2%. (c) The average saccade response time for Monkey G was 218 ± 32 ms, and the antisaccade response time 267 ± 24 ms, with a correct rate of 85.7 ± 4.6%. The data underlying the graphs shown in the figure can be found in https://osf.io/2q357/.

covariations of pupil diameter and CBV can be selectively observed in the SEF by manipulating reward and cognitive effort.

## Results

We recorded SEF activity by fUS imaging in 2 monkeys ($n$ = 26 sessions for Monkey S and $n$ = 20 sessions for Monkey G) trained to perform a pro/antisaccade task (Fig 1A). The task differs from the usual pro and antisaccade task where the information about the nature of the trial is conveyed by the fixation point and not by the peripheral target. We decided to train our animals in this variant so that future experiments parametrically manipulate the perceptual difficulty of target selection on each trial (by varying the ratio (height/width)). Both monkeys performed the task reliably across all recording sessions, and the average correct rate of both monkeys was approximately 85% (Fig 1B and 1C). The 2 monkeys showed significant shorter latencies for prosaccades than antisaccades, confirming a higher cognitive effort when an antisaccade was planned (Monkey S: 197 ± 15 ms for prosaccades, 262 ± 10 ms for antisaccades, $p$ = 7e-9; Monkey G: 218 ± 32 ms for prosaccades, 267 ± 24 for antisaccades, $p$ = 2e-4, using Wilcoxon's rank test). All trials were used for futher analysis regardless of their nature (prosaccades versus antisaccades, right versus left) or whether they were successful or not, unless stated otherwise.

## 1 – Pupil diameter covaries with supplementary eye field CBV at short time scale

We wished to investigate the relationship between pupil diameter and brain activity without any a priori choice concerning the activated area. To map those areas, we applied the generalized linear model (GLM) to the fUS data using pupil diameter as the input matrix. In total, 600 trials were used for these analyses. Each trial is separated from one another by a random interval of 3 to 4 s, this jitter allowing avoidance of monkey anticipation between trials. We constructed the input matrix by realigning all pupil diameters with the target presentation time using the peak value of the pupil diameter during the presentation stage. Indeed, the pupil diameter was higher for the first trials (in blue in Fig 2A–2E) than for the last (in red in Fig 2A–2E). We choose, for reproducible measurements, the pupil diameter during the trials as the pupil diameter at the median first local maximum dilation (at 0.8 s for Monkey S (Fig 2A) and 0.6 s for Monkey G (Fig 2E)). The highlighted pixels in Fig 2B–2F are those for which $p < 0.05$ (before Bonferroni correction), indicating the pixels highly correlated with the pupil

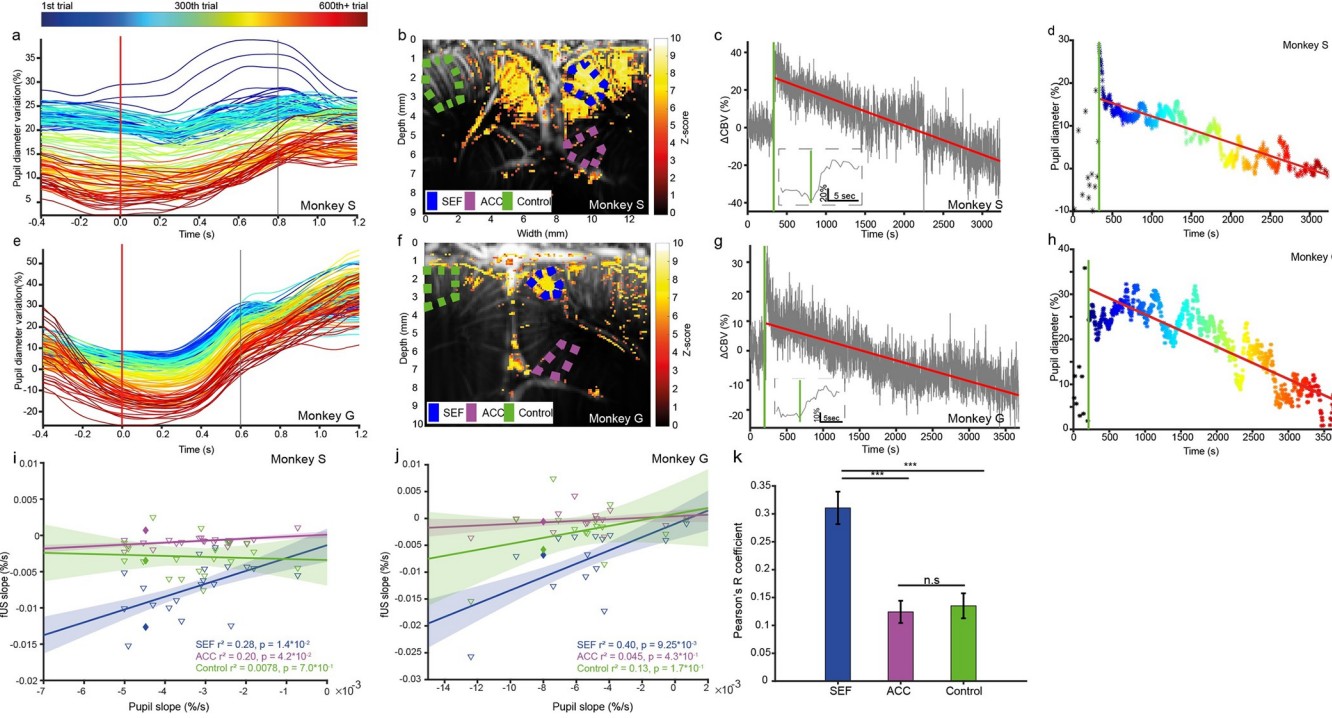

**Fig 2. Example of 1 session for each monkey of the vascular and pupillary responses and multiple sessions covariation of pupillary and vascular responses.** (a) Pupillary response over time. The color represents the number of the trial (blue: first trial, red: last trials), whereas the temporal abscissa represents the time prior to the presentation of the saccade or antisaccade cue. The vertical line represents the chosen time for the maximal dilation (0.8 s for Monkey S). (b) The vascular response of Monkey S using Fig 2D as an input matrix for the GLM. The background image consists of an anatomical image obtained by averaging all the Doppler films. The Z-score map was obtained using the GLM and thresholded using the Bonferroni correction ($p < 0.05$ uncorrected). (c) Example of a CBV response during the starting of a task for Monkey S., showing a step at the end of the baseline and the beginning of the task. The small square represents a zoom on the first trial. (d) Maximum pupillary dilation at 0.8 s after presentation of the cue using the same color code as in B. Black stars represent the pupillary diameter during baseline. (e) Same as for 2.a for Monkey G. The vertical line is at 0.6 s after presentation of the cue. (f) Same as for 2.b for Monkey G, using 2.h as an input matrix. (g) Same as for 2.c for Monkey G. (h) Same as for 2.d for Monkey G. (i and j) Scatter plot of the ΔCBV slope in function of pupil slope in SEF (blue), ACC (purple) et Control area (green) for both Monkey S (Fig 2I), and Monkey G. (Fig 2J), prediction intervals are the 95% prediction interval. The session represented in full lozenge is the session taken as example in a–d for Monkey S and e–h for Monkey G. (k) Pearson's correlation coefficient between pupil diameter (see d and h) and 3 areas in the brain: SEF (blue), ACC (purple), and Control (green) for all nonreward-modulated sessions for both animals ($n = 21$ sessions for Monkey S, $n = 13$ sessions for Monkey G), *** $p < 0.001$, ns: not significant. The data underlying the graphs shown in the figure can be found in https://osf.io/2q357/. All vertical green bars represent the end of the baseline and the start of the task. ACC, anterior cingulate cortex; CBV, cerebral blood volume; GLM, generalized linear model; SEF, supplementary eye field.

diameter in our cortical imaging plane. In these activated pixels, mostly consisting of the surface of the cortex, we found the activated area to be in the SEF, bilaterally for Monkey S and mostly in the left area for Monkey G. Such activation is consistent with the Paxinos atlas for the localization of the SEF. High correlations of the fUS signals with the pupil diameter were found in the SEF regions for both animals. Finally, we extracted the cerebral blood volume (CBV) temporal signal from our fUS data by spatially averaging the signal isolated in the functionally activated area (Fig 2C–2G). By looking at the very first trial of each task, we observed that the covariation between CBV and pupil signals occurred over short time scales, as both the fUS and pupil diameter signal exhibited a large and sharp increase (zoom in Fig 2C–2G). The CBV sharp increase is consistent with the results of the previous studies [22]. Finally, we can infer a similar activity in the pupil and in the SEF, as denoted by the application of the GLM.

### 2 – Pupil diameter covaries with supplementary eye field CBV at long time scale

We are now interested in knowing if the pupil diameter and the activity in the SEF is not only having covariation at short time scales, but also as long time scale, i.e., during a whole session.

During the successive trials of a nonreward-modulated prosaccade and antisaccade task, we observed a large and reproducible decrease in the relative CBV (rCBV) of the SEF, defined anatomically using the monkey brain atlas and functionally using the GLM, as previously described (Fig 2C–2G) (–23 ± 2%/h for Monkey S and –21 ± 5%/h for Monkey G) after the initial step induced by the start of the task (the first saccade) as described by Dizeux and colleagues [22]. Given the strong correlation between pupil diameter and cognitive engagement in the task, we examined the pupil diameter after a prosaccade or antisaccade task. We also observed a large and reproducible decrease in pupil diameter throughout the session (Fig 2D–2H, –9.3 ± 0.3%/h for Monkey S and –9.7 ± 3.1%/h for Monkey G) on a long time scale. The decrease in pupil diameter is correlated with the change in the activity of the SEF, as quantified next.

Because the step played an important part in the correlation between the pupil diameter and the Doppler signal in the SEF, we also performed an analysis on the decays without taking the step into account. We fitted the slope of the fUS signal (normalized by the baseline) in 3 regions: SEF, ACC, and Control, and the slope of the pupil during the task over different sessions. We then looked at the relationship between those slopes, by plotting the former as a function of the latter (Fig 2I for Monkey S, Fig 2J for Monkey G) and observed a relation between the slope of the pupil and the SEF ($R^2 = 0.28$ for Monkey S and $R^2 = 0.40$ for Monkey G). This relation is weaker for the ACC ($R^2 = 0.20$ for Monkey S, $R^2 = 0.045$ for Monkey G) and for the Control area ($R^2 = 0.0094$ for Monkey S and $R^2 = 0.13$ for Monkey G) indicating the relationship between the CBV in the SEF and the pupil even without the initial step.

Furthermore, we compared the squared Pearson's $R^2$ correlation coefficient between 3 regions of the brain (SEF, ACC, and a control area, anatomically chosen for the SEF and ACC and to be the non-activated cortical area farthest from the SEF) and the pupil signal. The $R^2$ was significantly higher (using a linear mixed-effects statistical model) in the SEF (0.31 ± 0.03, SEM) than in the ACC (0.12 ± 0.02, SEM) or control area (0.14 ± 0.02, SEM) (Fig 2K).

Those results seems to indicate that pupil activity, and SEF activity are linked together during the time of a session.

### 3 – Stronger vascular response in antisaccades compared to prosaccade trials in the SEF accompanied by larger pupil response

Given the correlation between the activity of the pupil and in the SEF, we were also interested in the difference of activity for a given cognitive load.

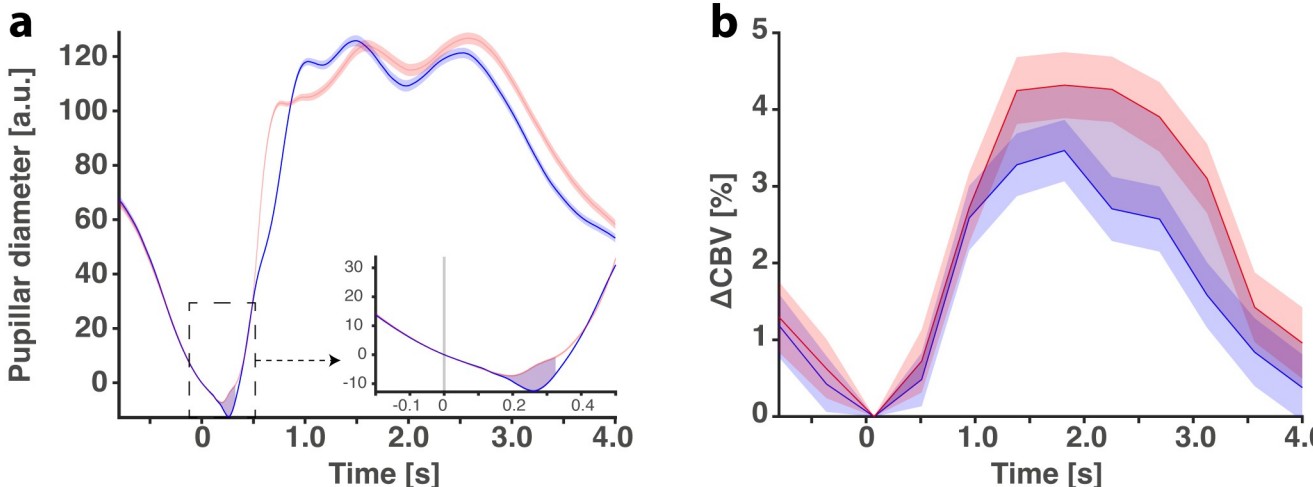

**Fig 3. Effect of cognitive effort (prosaccades versus antisaccades) on pupil diameter and cerebral blood flow.** (a) Pupillary diameter for prosaccade (in blue) and antisaccade (in red) +/− SEM averaged over all kept sessions in 2 animals. Purple area corresponds to the difference in the area under the curve one of both curves. (b) Same as (a) but for the ΔCBV. The data underlying the graphs shown in the figure can be found in https://osf.io/2q357/.

The successive trials consist of randomized prosaccades and antisaccades, as described in Fig 1. The cognitive effort required to perform an antisaccade is higher than the one required to perform a prosaccade. All sessions were kept for further analysis, without any discrimination between them. In those analyses, only the correctly performed pro or antisaccades were kept for analysis. Then, we analyzed the difference between the pupillary response for prosaccades and antisaccades. In 2015, Wang and colleagues [20] showed that the pupillary diameter is slightly bigger in preparation for antisaccades than it is for prosaccades. Here, we showed, with the same principle, that pupillary diameter is slightly larger for antisaccades than it is for prosaccades, even without a gap between the cue presentation and the realization of the saccade ($p < 0.001$). We measured the area between the antisaccade curve (Fig 3A, in red) and the prosaccade curve (Fig 3A, in blue) to quantify those slight differences in the pupillary diameter ($n = 32$ sessions), between 0 ms and 320 ms, as it corresponds to the median onset time for antisaccades plus the fixation of the targeted region of interest on the screen. A quite important hypothesis was to test the difference between pro- and antisaccades in the ΔCBV of the SEF to see if this area is not only responding to long-term monitoring of motivation and effort, but also short-term effort. We have realigned fUS signals on target presentations (Fig 3B, blue for prosaccades, red for antisaccades) to see if the SEF had a different vascular response for pro and antisaccades. We showed, here, that the vascular response for antisaccades is higher than the vascular response induced by prosaccades ($p < 0.01$, integration of the signal between 0 s and 4 s as it corresponds to the time for estimated HRF to return to the baseline), showing that the SEF is not only sensitive to long-term variations (minute variations) but also short-term effort-related variations (single trial variations).

## 4—Reward magnitude modulates both SEF activity and pupil diameter

Since correlation does not imply causation, we wanted to manipulate reward to test the causality of our measures. We measured engagement in the task by slightly adapting our paradigm by adding a color code at the fixation point on each given block of 100 trials and modifying the magnitude of the reward. The potential reward delivered for each correct pro or antisaccade was red for 0.5 reward units, blue for 1.0 reward unit, and green for 1.5 reward units (Fig 4A

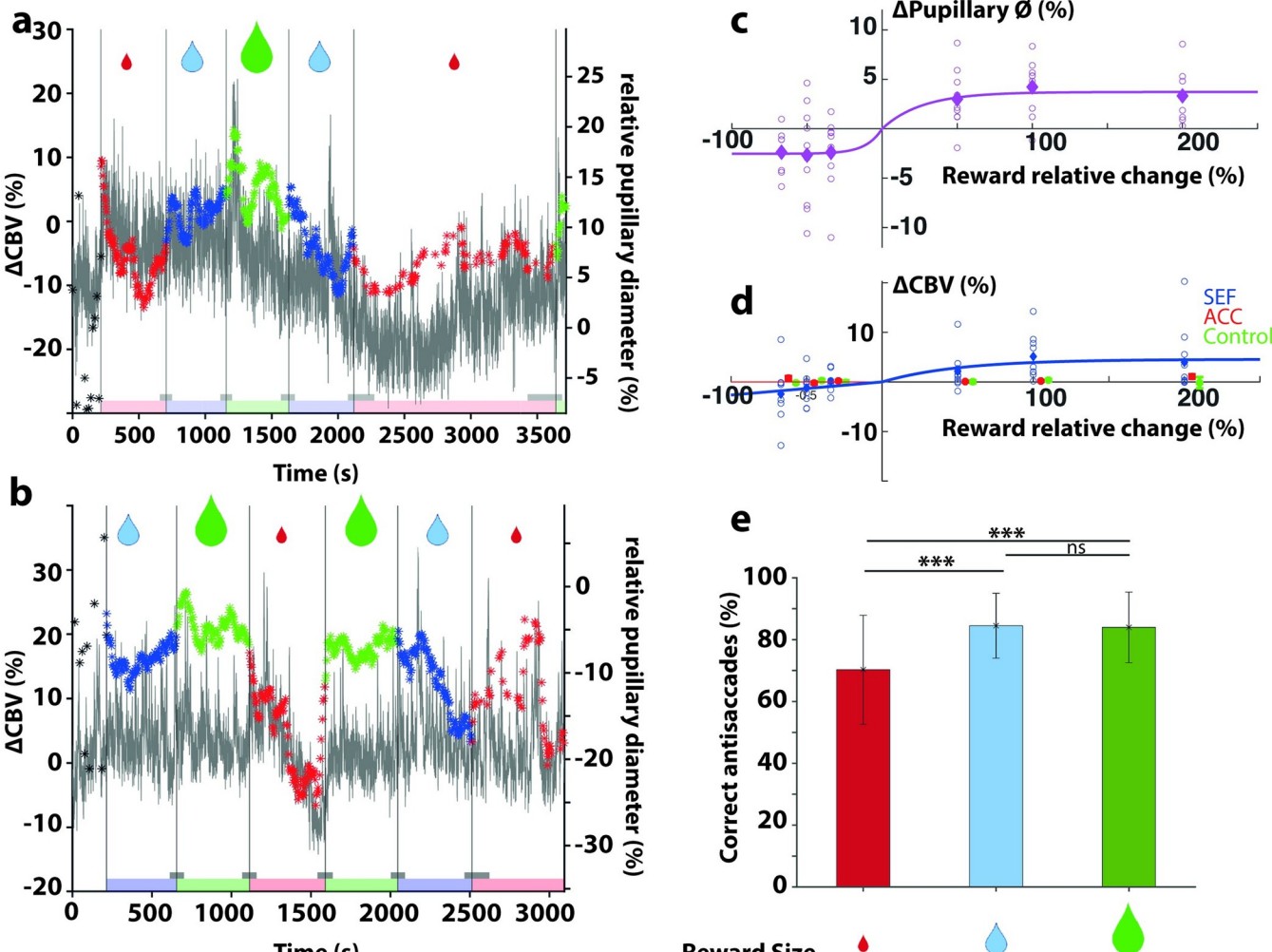

**Fig 4. Reward modulation during the task for both Monkey S and Monkey G.** (a) CBV and pupillary response over 1 session for Monkey S. The colors represent the quantity of reward obtained after a successful trial (green: 1.5× the base reward, blue: 1× the base reward, and red: 0.5× the base reward). (b) Same as for (a) for Monkey G. (c) Pupil dilation according to the change in reward for Monkeys S and G. Mean ± standard error of the mean. (d) The CBV changed according to the change in reward, all sessions for Monkeys S and G. Blue represent the SEF, red the ACC, and green the control area. Mean ± standard error of the mean. (e) Animal disengagement during the task for antisaccades for low (red), normal (blue), and high (green) reward. Mean ± standard error of the mean. ns: not significant, *** $p < 0.001$. The data underlying the graphs shown in the figure can be found in https://osf.io/2q357/. ACC, anterior cingulate cortex; CBV, cerebral blood volume; SEF, supplementary eye field.

and 4B). During the task, the ΔCBV changed during the transition from one reward level to another, as did pupil dilation. We also observed a slight disengagement of monkeys when the task is high-cognitive demanding (e.g., antisaccade) for a low reward (e.g., 0.5 reward unit for red fixation point). The observed saturation for pupil size and CBV for the largest reward changes may reflect the limited delta of the subjective perception of reward manipulated in our experiments.

We wanted to quantify pupil dilation and the ΔCBV during such transitions. We computed the transition between the 2 levels of reward by computing the average CBV and pupil diameter for 10 trials before the transition and 10 trials after. The difference between the values after versus before the transition gives the increase or decrease induced by the transition. Pupil dilation decreased in the transition from a higher to lower reward and increased in the transition

from a lower to higher reward (Fig 4C). We observed an increase in the ΔCBV in the SEF during a transition from a lower to higher reward, but no statistically significant measure was obtained for a transition from higher to lower reward (Fig 4D, in blue). We did not identify any significant difference in covariations between the different reward manipulations.

Interestingly, we observed a marginal effect on the ACC (Fig 4D, in red) for the reward transition from 0.5 units to 1.5 units and 1.5 units to 0.5 units. More rostral investigations of the ACC would be required to measure such effects in this area. A control area showed no augmentation or decrease (Fig 4D, in green). Finally, increasing the reward resulted in augmentation of both pupil dilation (Fig 4C) and ΔCBV in the SEF (Fig 4D, blue) but also a modification of the engagement to perform the task (Fig 4E): Animals were less engaged for the antisaccades in the task during low rewarded block than for normal or highly rewarded blocks (56.2 ± 4.4% and 85.3 ± 3.7% for Monkey S and G, respectively, compared to 77.1 ± 2.9% and 91.1 ± 2.3% for a normal reward for monkey S and G, respectively), indicating that our manipulation of reward was accounted by the animals to adapt their behavior according to the trial difficulty. Between blocks of distinct reward values, we did not observe significant changes in covariations between pupil and fUS responses in SEF. The duration of the periods (only 100 trials) during reward manipulation may not allow us to quantify such modulations observed at the session length (approximately 600 trials). Overall, at a transient time scale (few trials), the fUS activity in the SEF was strongly modulated in real time by the reward, as was pupil diameter.

## Discussion

We combined pupil signals and real-time fUS imaging of NHP during cognitive tasks, which allowed us to infer localized CBV responses within a restricted part of the dorsomedial prefrontal cortex, referred to as the SEF, an area in which antisaccade preparation activity is also recorded. Inversely, SEF neurovascular activity measured by fUS imaging was found to be a robust predictor of specific variations in pupil diameter over short and long time scales. The manipulation of reward and cognitive efforts performed by the animals resulted in strong temporal covariation of pupil diameter and CBV within the SEF. Overall, these results show the region of the SEF to be an underestimated pivotal element within the medial frontal cortex of primates for monitoring and implementing the cognitive effort signals observed within autonomous networks.

In previous studies, SEF neurons have been shown to participate in the selection of eye movements by representing the context-dependent action value of various possible oculomotor behaviors [24]. However, the SEF alone does not have the aptitude to directly select eye movements [24]. In the same vein, the SEF does not directly participate in the rapid inhibitory control of eye movements in response to sudden changes in task requirements.

In both human and nonhuman primates, seminal studies have shown that SEF can indirectly influence the selection of behavior through modulation of this primary selection process in the motor structures. In humans, numerous studies suggest a role for the SEF in controlling "internally generated" eye movements during the performance of complex learned behavior [25–27].

Rare studies have been capable of further examination. Some reports from sporadic patients with focal lesions of SEF have shown great difficulty for these patients across a range of saccadic tasks [25–28], as well as changing from an initial saccade plan to an alternative one [29,30]. These results were confirmed by functional imaging of healthy individuals performing a change-of-plan saccadic task, resulting in enhanced SEF activity when subjects successfully changed their saccadic plans compared to following a predictive plan [31]. Finally, some

studies in human neuroimaging experiments have related these voluntary controls of saccade plan to a more general influence in the context of speed-accuracy tradeoff experiments [32,33].

Based on these findings, it has been proposed that a significant role of the SEF lies in implementing control over conflicting internally generated saccadic plans [30,31]. In macaques, SEF also shows systematic changes in activity during learning of new stimulus-response associations [32].

Even after the task set has been learned, monitoring of behavior is necessary to catch changes in the environment or possible mistakes due to response conflict or inadequate attention that guides their behavior as long as they are motivated to do so [33–36]. Our results reinforce these interpretations of the role of SEF in the monitoring of behavior. However, they may have extra critical implications because variations in pupil diameter have been observed for various tasks [1–5]. Two principal explanations have been provided to account for such pupil-effort covariation. First, a direct "bottom-up" influence on decisions produces a bias toward accepting an effort. This would be consistent with the widely held view that the strength of neural representations for choice attributes directly influence the decision. For example, it has been shown that intensifying encoded rewards through the simulation of future episodic events is linked with decisions that promote higher long-term payoffs and even increase prosocial behavior.

As for neural implementation, phasic locus coeruleus (LC) activity is known to transmit feedforward information to the SEF via ascending projections to the prefrontal cortex (PFC), providing a plausible pathway for such a bottom-up influence. Recent work has shown that LC neurons can reflect both cognitive and physical efforts with a subsecond precision [37].

Therefore, the neural readout of the autonomous activation associated with arousal could provide an additional mechanism by which the arousal signal observed here may bias choices, serving as a signal that the organism is indeed ready to accept the physical challenge.

In the ACC, unlike the SEF, there was not even a tendency of heightened CBV modulation under conditions of cognitive effort. This finding is compatible with an earlier report showing that ACC neurons in the monkey are not selectively active during the countermanding of saccades, an operation assumed to involve cognitive effort and inhibition of action [38]. However, it stands in sharp contrast to a large body of literature, based on functional MRI imaging in humans, indicating that activation in the ACC is strongly heightened under conditions of effort [39–42]. Here, we report that during a sustained execution of a demanding oculomotor task over more than 1 h, the neurovascular coupling (as assessed by fast ultrasound imaging, fUS) in the SEF of macaques, progressively drops with time, correlating with a simultaneous drop in pupil size and performance. When changing reward ratio, intrinsic brain mechanisms might compensate for the progressive drop in cognitive performance by instantiating alternations of high and low performance. These variations in behavioral performance might be phase-locked to variations in attention- and perception-related information as shown in macaque FEF neuronal populations [43].

There are several possible explanations for this discrepancy. There may be a species-specific difference, such that neurons in the human ACC monitor cognitive effort, whereas those in the monkey ACC do not. This seems improbable because, in general, anatomically homologous areas appear to serve similar functions in the 2 species [44]. This cannot, however, be altogether ruled out. The human ACC possesses a cell type not found in the monkey ACC [45] and, therefore, may serve a function not served by the monkey ACC. It is possible that our recording sites lay outside the region of the ACC responsible for effort-related activity [46]. This also seems improbable because we deliberately recorded in the subregion that is connected to the SEF [47] and in which, accordingly, it would be most reasonable to expect to find activity sensitive to cognitive effort in an oculomotor task. It is also possible that the cognitive-

specific bold signals detected in human fMRI studies are related to neural events other than spiking activity and CBV, for example, presynaptic potentials [48]. It may be the case that ACC activity, even in humans, does not exhibit enhanced spiking activity under conditions of cognitive effort. For the ACC to serve a cognitive effort and alert the rest of the cortex to the presence of cognitive effort would require enhanced spiking activity because spikes are the currency used between the ACC and other cortical areas. Thus, the remaining conclusion is that the ACC does not monitor cognitive effort.

Overall, our observations are consistent with a possible top-down influence from the SEF to the noradrenaline arousal system, which may serve to transmit information about the commitment to overcome a great physical demand, thus resulting in automatic accelerating upregulation of arousal states to prepare the organism for the upcoming challenge associated with the recent choice. As SEF activity and pupil are strongly correlated over large time scales, our results also allow us to conclude that within the medial frontal cortex of primates, aside from the ACC, the SEF may also play a role in the implementation of the arousal signals observed within autonomous networks.

## Methods

### Animal model and behavioral data

All experimental procedures were designed in association with the veterinarians of the ICM Brain and Spine Institute, approved by the Regional Ethical Committee for Animal Experiment (CREEA IDF n˚3, agreement number A-75-13-19; Ministère de l'Education, de l'Enseignement Supérieur et de la Recherche under the project reference APAFIS #6355–2016080911065046), and performed in compliance with the European Community Council Directives (86/609/EEC). Functional data were acquired from 2 captive-born rhesus monkeys (*Macaca mulatta*), S and G, trained to perform various types of visual tasks. In the saccade task, the animal has to fix its gaze on the cue object presented on the right or left side of the screen; in the antisaccade task, it has to fix its gaze on the opposite side from where the cue appeared. Each animal performed at baseline (200 to 220 s, random) followed by saccades and antisaccades (randomized) over 1 h. During data acquisition, the eye position of the primate was monitored at 1 kHz using an infrared video eye tracker (Eyelink 1k, SR-Research), which enabled live control of the behavioral paradigm and the delivery of a reward based on the success or failure of a visual task [49].

### Experimental setup

We recorded 46 sessions (26 for Monkey S and 20 for Monkey G) of a variant of prosaccade and antisaccade tasks [50], with 2 kinds of sessions (Fig 1).

The conventional session, without reward modulation, consisted of only a blue square before the prosaccade or antisaccade cue, and the reward was kept constant within and between all sessions. For the second type of task, with reward modulation, the same basal reward was retained and the animal was presented with 3 colored dots (red for 0.5 reward unit, blue for 1 reward unit, and green for 1.5 reward unit) before the prosaccade or antisaccade cue. Behavioral data, such as pupil diameter, were recorded with an EyeLink system and CBV using a functional ultrasound scanner for all sessions.

All tasks were driven by EventIDE software (OkazoLab, the Netherlands).

The reward was calibrated to the weight of the primate and the model of the rewarding tube (approximately 30 ms/kg for the electronic valve), which delivered sugary water. Primates were under mild fluid restriction (approximately 30 mL/kg/day) and could drink ad libitum while working.

## Implant and probe for functional ultrasound imaging for awake cooperative monkeys

The head of the monkey was fixed using a surgically implanted titanium head post (Crist Instrument, Maryland, United States of America). After behavioral training of the animals, a recording chamber (CILUX chamber, Crist Instrument, Maryland, USA) was implanted, and a craniotomy (diameter 19 mm) was performed (mediolateral: +0 mm, antero-posterior: +26 mm). The ultrasonic probe with sterile ultrasonic gel was then inserted into the chamber.

## Functional ultrasound (fUS) recording

Changes in CBV were measured using a real-time functional ultrasound scanner prototype (Iconeus and Inserm U1273, Paris, France) with a custom linear probe (128 elements, 15 MHz, $100 \times 100\ \mu m^2$ of spatial resolution). The probe was positioned in the recording chamber using an home made adapter (represented in black on the image on the left) with a small notch allowing the probe to be positioned at the same position for each recording session. Moreover, the thickness of the imaging plane (from 700 μm at the start to 400 μm at the focal plane, around 8 mm from the probe) allowed a repeatable field of view even though the position was slightly variable. This scheme is now presented in S1 Fig. Data were acquired by emitting continuous groups of 11 planar ultrasonic waves tilted at angles varying from −10˚ to 10˚. Ultrasonic echoes were summed to create a single compound image acquired every 2 ms. Final Doppler images were created by averaging 200 compound ultrasonic images after spatiotemporal filtering based on the singular value decomposition of the ultrasonic images. The acquired images have a pixel diameter of $100 \times 100$ μm and a slice thickness of 400 μm. The CBV is acquired continuously at the rate of 2.5 Hz and extracted at each trial on a time window of −1.2 to 4.4 s (from target onset). Those values were chosen as they are a multiple of 0.4 s, which is the sampling of the fUS imaging.

## Eye movements and pupil recordings

Eye movements and pupil diameter were recorded during the tasks using a video eye tracker (Eyelink 1k, SR-Research) connected to an analog-to-digital converter (Plexon, Texas, USA). All data were collected using Plexon software and analyzed using MATLAB (MathWorks, Massachusetts, USA). Saccades were detected when the eye's horizontal velocity went over $30˚\ s^{-1}$.

## Data processing

**Generalized linear model.** Doppler data were analyzed using a GLM approach implemented in Matlab. The stimulation pattern in the design matrix was convoluted with the fUS-determined HRF and a Z-score and CBV change map were obtained. The activation maps show the Z-score of all pixels in the images with a $p$-value $< 0.05$ (before Bonferroni correction). We chose the region of interest (ROI) within the SEF based on the Z-score map and Paxinos atlas for macaque brains and the signal was averaged to obtain a single temporal signal. The spatially averaged signal was then expressed as the relative increase in CBV (in percent) by subtracting the baseline CBV (calculated during the baseline at the beginning of an acquisition) normalized to the baseline CBV.

**Determination of the pupil diameter.** The pupil diameter was expressed in percent by subtracting the baseline value and then dividing the difference by the baseline value (in which we excluded all blinks and moments in which the eyes were closed). We then determined the maximum dilation diameter following a task by realigning the pupil diameter at the onset of

the cue presentation. We chose the first local maximum of the pupil diameter (0.8 s after target onset for Monkey S and 0.6 s after target onset for Monkey G) to extract the pupil diameter for the ith trial.

**Fitting of the hemodynamic response.** The hemodynamic response was determined by averaging the CBV response of all trials and fitting the average by an inverse-gamma probability distribution using MATLAB *lsqcurvefit* (Optimization Toolbox) algorithm for least square nonlinear fitting, as previously described by other authors [51].

**Statistical analysis of the hemodynamic responses.** Statistical analysis between 2 groups was performed using the Wilcoxon rank test, due to the non-normality of our data, using the Matlab ranksum function, the null hypothesis being no statistical difference between the 2 groups. If more than 2 groups were available and the data hierarchically organized, we used a linear mixed statistical model. Data was homogenized using a square root transformation and the variance of homogeneity assessed using the Bartlett test and residual normality the Shapiro–Wilk test.

## Supporting information

**S1 Fig. Custom Holder for fUS recording and average of all sessions for the ΔCBV in the SEF, ACC, and control area for the 2 animals.** (a) Custom holder for adaptation of the ultrasonic probe (15 MHz) to the recording chamber on the animal. (b) ΔCBV for the SEF (in blue), the ACC (in red), and the control area (in green) +/− SEM across all sessions. The data underlying the graphs shown in the figure can be found in https://osf.io/2q357/. ACC, anterior cingulate cortex; CBV, cerebral blood volume; fUS, functional ultrasound; SEF, supplementary eye field.
(TIF)

**S2 Fig. Average trial response in function of the reward for the ΔCBV and the pupil for the 2 animals.** (a) Average ΔCBV response for low (red), medium (blue), and high (green) reward. (b) Integration between t = 0 s and t = 4.4 s of the previous ΔCBV curve. (c) and (d) Same for the pupil diameter. Integration is calculated between t = 0 ms and t = 320 ms. n.s.: not significant, *** $p < 0.001$ The data underlying the graphs shown in the figure can be found in https://osf.io/2q357/
(TIF)

## Acknowledgments

Part of this work was carried out on the Phenoparc core facility of ICM. We gratefully acknowledge Morgane Weissenburger and Estelle Chavret-Reculon for the day to day management of the animal facility and Bénédicte Daboval and Lucile Aubrée for the veterinary care.

## Author Contributions

**Conceptualization:** Fabrice Arcizet, Thomas Deffieux, Mickael Tanter, Pierre Pouget.

**Data curation:** Julien Claron, Julie Royo, Thomas Deffieux, Mickael Tanter.

**Formal analysis:** Julien Claron, Thomas Deffieux, Mickael Tanter.

**Funding acquisition:** Mickael Tanter, Pierre Pouget.

**Investigation:** Julien Claron, Julie Royo, Fabrice Arcizet, Thomas Deffieux, Pierre Pouget.

**Methodology:** Julien Claron, Thomas Deffieux, Pierre Pouget.

**Project administration:** Thomas Deffieux, Mickael Tanter, Pierre Pouget.

**Supervision:** Pierre Pouget.

**Validation:** Fabrice Arcizet, Thomas Deffieux, Pierre Pouget.

**Visualization:** Julien Claron, Fabrice Arcizet.

**Writing – original draft:** Julien Claron, Julie Royo, Fabrice Arcizet, Thomas Deffieux, Mickael Tanter, Pierre Pouget.

**Writing – review & editing:** Julien Claron, Thomas Deffieux, Mickael Tanter, Pierre Pouget.

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
