## [Editor Report · Decision Letter 0]

17 Aug 2021

Dear Pierre, 

Thank you for submitting your manuscript entitled "The Supplementary Eye Field Tracks Cognitive Efforts" for consideration as a Research Article by PLOS Biology.

Your manuscript has now been evaluated by the PLOS Biology editorial staff, as well as by an academic editor with relevant expertise, and I am writing to let you know that we would like to send your submission out for external peer review. Please accept my apologies for the delay in sending this decision to you.

Please re-submit your manuscript within two working days, i.e. by Aug 19 2021 11:59PM.

Kind regards,

Gabriel Gasque

Senior Editor

PLOS Biology

ggasque@plos.org

---

## [Decision Letter · Decision Letter 1]

19 Oct 2021

Dear Pierre,

Thank you for submitting your manuscript "The Supplementary Eye Field Tracks Cognitive Efforts" for consideration as a Research Article at PLOS Biology. Your manuscript has been evaluated by the PLOS Biology editors, by an Academic Editor with relevant expertise, and by four independent reviewers. Please accept my apologies for the long delay in sending the decision below to you.

In light of the reviews (below), we will not be able to accept the current version of the manuscript, but we would welcome re-submission of a much-revised version that takes into account the reviewers' comments. We cannot make any decision about publication until we have seen the revised manuscript and your response to the reviewers' comments. Your revised manuscript is also likely to be sent for further evaluation by the reviewers.

We expect to receive your revised manuscript within 3 months. 

**IMPORTANT - SUBMITTING YOUR REVISION**

*Re-submission Checklist*

*Published Peer Review*

*PLOS Data Policy*

*Blot and Gel Data Policy*

Sincerely,

Gabriel Gasque

Senior Editor

PLOS Biology

ggasque@plos.org

REVIEWS:

Reviewer #1: In this study, Claron et al. used functional ultrasound imaging while measuring the pupil diameter in macaque monkeys performing a saccade task to investigate the link between SEF activity and cognitive effort. 

The paper is interesting, well written and overall pointing out a relationship between SEF and cognitive effort. However, the quality of the manuscript could be greatly improved by adressing several concerns.

1. Further clarifications are needed regarding the data analysis. It is unclear why the values of pupillary diameter were chosen at 0.8s and 0.6s for the data analysis in Fig.2; for instance, why not using 0.8s for monkey G, since the diameter is higher than at 0.6s ? Additionally, the temporal profile of the pupil size variation is different during the first trials as compared to the last trials in monkey S, making it difficult to chose a consistent time point. Therefore, these time points seem quite arbitrary and should be more justified, especially since it may have an important impact on the results. 

2. In this regard, the interest of using only the pupil diameter at maximum dilation for further analyses is not clear to me and should be better explained. Is this step used for all results depicted in Fig 2, and if yes, why not using all pupil diameter values (after temporal realignment and resampling with the fUS signal) ? This would enable to study the correlation between cognitive effort and SEF activity at shorter time scales occuring in a single trial (which is something discussed later but only in terms of difference between antisaccade and prosaccade: as variations of pupil size are also occuring at the trial level, it would be more convincing to take them into account).

3. It would be valuable to show more results at the group level instead of examples of one session for each monkey. The supplementary figure is useful and should also include the average of all sessions for the pupil diameter variations. The variability across the Z-score maps for the different sessions should also be shown or discussed. 

4. In Figure 3, the same time scales should be used for pupil diameter and CBV in order to visualize the full temporal profiles of the responses and discuss the correlation between them at the trial level. 

5. How was performed the positioning of the probe in the cranial window to select a repeatable field of view ?

Reviewer #2: In this paper entitled 'The supplementary eye field tracks cognitive efforts', Claron et al. use fast ultrasound imaging to monitor in real-time the hemodynamic response evoked in the supplementary eye field (SEF) during the execution of a demanding cognitive task. Pupil size has been shown to track cognitive effort both at the single trial level and at the session level of such tasks. The authors thus seek to characterize the relationship between pupil size and SEF cerebral blood volume variations (CBV). They show that SEF hemodynamic activity is a robust predictor of pupil size variations at multiple time-scales including when experimentally manipulating cognitive effort and reward expectation. 

The data is compelling and quite surprisingly shows that this correlation between CBV and pupil size is restricted to the SEF and does not extend into the adjacent cingulate cortex or along the adjacent cortical convexity. While this brings a very novel understanding of the physiological underpinnings of cognitive effort, it also raises a lot of crucial questions for upcoming research projects. 

I thus strongly recommend the publication of this work once my comments below have been addressed (page and line numbers would have been useful to the reviewing process):

Major:

1. Abstract: the authors use in their concluding statement the expression 'SEF is an underestimated but pivotal cortical area for the monitoring and implementation of cognitive effort signals': I strongly recommend the authors to reformulate as this suggests a causal role, which has actually not been demonstrated in the study. More generally speaking, I recommend the abstract to be reworked in depth as it does not make justice to the results: key concepts (e.g. cognitive effort, CBV) are not defined and the text can be clarified (e.g. 'Here, we tested […] is also recorded' appears like the same information comes up multiple times in multiple sentences). 

2. I recommend the authors also scrutinize the introduction in order to enhance the impact of their work. For example: 1) '..; which mediates arousal, are likely involved': references should be added; 2) Second paragraph 'In principle … Alternatively ….', please reformulate and back up with strong references; 3) 'Although the function …': I think this statement is extremely vague. The authors use anatomical evidence to suggest a role of SEF activity in the control of pupil size. This should be made explicit (control? Single controller? Why not FEF as a controller? Why not ACC?); 4) 'suggesting valid qualities': please reformulate, I am not sure I get the actual meaning here; 5) 'Given that the SEF is known …': I am missing the logic link here: why is the fact that the SEF is crucial to anti-saccades calls to investigate pupil size in this context? This should be made explicit. 

3. CBV is extracted "using a random interval of 3 to 4 s after each trial": why random? Please describe trial length statistics and reward timing. It might be useful to increment the GLM with variables of non interest at this stage, such as trial outcome (i.e. reward / no reward or trial type anti- pro-saccade). 

4. In section 3 of the results, better describe how many sessions were excluded per monkey. Also, rather than using an arbitrary threshold of 0.6 you might want to consider using IQR.

5. Section 3 is conceptually very important. What would further strengthen it is 1) to produce the equivalent of figures 2i,j,k for pro- and anti-saccades independently and 2) compare the strength of correlation between CBV and pupil for anti- and pro-saccades across sessions. The idea is to test whether correlation is strictly equivalent or is stronger in high demand trials relative to lower demand trials. 

6. Can the authors produce the plots of figure 4c,d for performance and the plots of 4e for CBV and pupil size as the information provided by each is different. In this context, can the authors say something about the observed saturation for both pupil size and CBV as a function of reward change. Is the coding of relative reward only qualitative? Last, can the authors quantify the correlation between the delta pupil size and the delta CBV across all reward transitions and across all sessions. This would strengthen their point, all the more that the single example shown in figure 4a,b does not suggest a strict correspondence as illustrated in figure 2 in the absence of reward manipulation. The underlying question is whether the correlation between SEF CBV and pupil size is comparable across all modelled variables or whether correlation is stronger for some of them? This would help better qualify the mechanism under consideration. 

7. Can the authors speculate on why CBV and pupil size decrease linearly when reward is constant (figure 2c,g) but not when it is not (figure 4a,b) -I apologize in advance if I missed some methodological detail here. 

8. Does covariation between SEF CBV and pupil size change between correct and incorrect trials? If this is the case, this would be quite interesting functionally speaking. 

9. The discussion includes an extensive discussion of human ACC. A recap of human SEF activity in effort and motivational tasks would be useful. 

10. At the end of the discussion, there is a mention that CBV changes precede pupil size changes. I apologize upfront if I missed this analysis, but I couldn't find it in the manuscript. This is obviously a very useful and important information. Can you please clarify. If this is indeed the case that CBV precedes pupil size change, and depending of the latency between the two signals, the authors will want to explain if and how this impacts the reward manipulation section, as the data of both signals is sampled on the same number of trials around the time of shift in in reward size. 

11. Overall, the manuscript should be re-read thoroughly for English phrasing and fluidity. Generally speaking, the impact of the work can be strengthened by initiating every result section by a statement of the question being asked (this often comes quite late in the section), and a conclusion statement wrapping out what has been demonstrated. 

Minor:

1- Introduction, second paragraph: replace short memory by short-term memory

2- Results, title 1: I sugget something more specific for example 'Pupil size covaries with SEF CBV at short time scales'

3- Figure 1, indicate time scale in panel a for pupil diameter and rCBV

4- Did the authors mean "after" rather than before Bonferroni correction in first paragraph of section 1 and elswhere? If not, then this is a pb

5- Results: "using a random interval of 3 to 4 s after each trial ONSET"

6- Results: monkey G. (Fig. 2.e) not Fig. 2.d

7- Figure 2: panels I and j : indicate monkey identity + indicate p of correlation

8- Results, title 2: I suggest something in the line of 'Pupil size covaries with SEF CBV at the time scale of the session'

9- Results '… after the initial step': please specify

10- Results 'The decrease (no s) in pupil size correlated with the change in the activity of the SEF. THIS IS QUANTIFIED NEXT'

11- Results, I would place the description of the behavioral performance of the monkeys on the pro- and anti-saccades under section 3. Because that's where it is relevant. At the beginning of the results, you might consider describing more general behavioral features such as success rate.

12- Results, section 3: 'even without preparation': please clarify what you mean by this

13- Results, section 4: 'Reward magnitude modulates BOTH SEF activity and pupil diameter

14- I would suggest to uniformly use pupil size or diameter but not both. 

15- The authors will want to use the word CUE instead of hint. 

Reviewer #3: In this manuscript, the authors address the role that the supplementary eye fields (SEF) may play in relating to pupillary responses during a cognitively demanding task. The work is done in macaque monkeys, which is an important animal model to help establish the underlying neural mechanisms linking cognitively demanding tasks to pupil dilation, which is commonly observed in humans. Neural activity in the SEF is inferred by measures of cerebral blood volume (CBV) via functional ultrasound, a technique in which the authors are world leaders. Measuring SEF activity via CBV is akin to functional neuroimaging in some ways, with the exception that it comes with a better temporal resolution. The authors also manipulate reward levels to further examine the relationship between SEF-CBV and pupil dilation.

There are a number of positives to the manuscript. The experiments were conducted with great skill, and the authors present largely common trends across two animals. Further, the authors identify of a gap in knowledge in the literature, and have found interesting distinctions between what was found in the SEF (which did relate to the pupil) versus the nearby anterior cingulate cortex (which did not). The paper is also largely well written, although I do have a number of comments below which I think may help convey the core message more clearly. 

Main points

While I really enjoyed this manuscript, I struggled with whether it is fundamentally about establishing the relationship between SEF-CBV activity and pupillary responses (implied in the abstract) or about linking SEF activity to cognitive effort (implied by the title). My view is that the paper is more strongly positioned as the former. Further, the notion of how cognitive effort can be dissociated from other related phenomena, like attention or motivation, is complicated, and I am not convinced that the manipulations of reward used here do indeed dissociate effort from attention/motivation. If the authors agree with my sense that the paper is about establishing the SEF as an additional frontal area influencing the pupil, then aspects of the title and introduction could change (e.g., the reward manipulation becomes another way to test the link between SEF activity and the pupil). Alternatively, if the authors feel that the paper is about cognitive effort, then I think they need to do a better job rationalizing the additional value of pupil measurements. 

The authors do a good job establishing short- and long-term correlations between SEF-CBV values and the pupil. For the short-term correlations however, and specifically regarding the fact that higher SEF-CBV values and greater pupil dilation accompany anti-saccades, I do worry that these results may reflect a confound of more saccades made on anti- vs pro-saccade trials, due to the inclusion of error trials. Is it possible to redo the analyses shown in Fig 3 using only correctly-performed anti-saccade trials? 

In the variant of the anti-saccade trial used here, the orientation of the peripheral target conveys the instruction to generate a pro- or anti-saccade. This variant differs from that used both by the Schlags in their seminal SEF work, as well as by the Wang studies in humans, where a feature of the initial fixation point conveyed task instruction. The latter variant introduces a greater separation in time of task preparation (which precedes target onset) from the saccadic response. I was curious as to the rationale for the authors to choose the variant of the anti-saccade task that they did. I think it would also help readers if the authors were more explicit about the difference in their task from that used by the work cited previously (the rationale and distinctions from previous work could go in the first paragraph of Results).

Upon first reading, I struggled a bit with aspects of Figure 2. Regarding panels 2c/g and 2d/h, it would help if the authors could convey when the task actually started, and whether this coincided with changes in luminance/overall illumination. Am I correct in presuming that the task started at around the 300s point? Also, regarding panels 2i/j/k, please specify in the results narrative that these are no longer single-session data, but instead summarize data across a larger sample of days. This information is currently only available deep in the Figure legend for Fig 2 (as an aside, showing data across multiple experimental session would seem to contradict the title to the Figure 2 legend, which emphasizes that the data is single-session data; this could also be corrected). 

In the second paragraph of the discussion, the authors reference "two principal explanations" for their data. The first explanation follows in the next sentence, but I was not clear about the second explanation. Indeed the next few paragraphs would seem to be about qualifying the bottom-up explanation (which I presume is the first "principal explanation"), and the absence of increased CBV in the ACC. I could not find the second explanation; is it the top-down influence mentioned in the final paragraph? Regardless, please clarify these explanations in the Discussion. 

Minor points

It is admittedly a stylistic point, but I felt there were aspects of abstract and introduction that were perhaps overstated. Notions like "underestimated yet pivotal" may be stretching thing a bit, and one could argue that the "pivotal" nature hasn't been established in this manuscript, since there was no true causal manipulation via stimulation or lesioning. Further, while I agree it is important to address the role of the SEF in pupil responses, I don't think many would be overly surprised that SEF activity relates to the pupil (given results from the FEF and ACC); others perhaps simply haven't measured it. Similar concerns about "pivot" relate to the usage of terms like "critically involved"; these strike me as terms that are just too strong to apply for the current dataset. 

Please define CBV on first use in the abstract

3rd paragraph of Introduction "…suggesting valid qualities for ongoing…". Not sure "qualities" is the right word here. Do the authors mean that pupil dynamics can provide a "proxy for" or "reflection of" cortical processing? 

Please consider adding references to: 1) The first paragraph of intro where it states that the dmPFC mediates arousal. 2) Second paragraph of intro; consider references for the SEF projections to the oculomotor nucleus, and references for how the ACC/SEF/FEF may modulate the olivary pretectal nucleus. 

Reviewer #4: In an excellent paper, Claron and colleagues report results of a pupillometry and fUS study of macaca mulatta cortex activity while monkeys perform in a rewarded saccade / anti-saccade task. The authors show fairly convincingly that SEF activity correlates more strongly than do ACC and control regions with pupil dilation (otherwise linked with cognitive effort), and that both SEF signal and pupil diameter covary reliably with reward magnitude, cross-session dynamics, and response to anti- vs pro-saccade trials and errors - all hallmarks of cognitive effort.

On the basis of these correlations, the Authors argue that SEF activity tracks cognitive effort, yet additional tests might provide stronger evidence. Presumably, for example, if animals exert more effort on a given trial, they would be more accurate and faster, or (better yet), faster controlling for accuracy and vice versa. As such, it would be nice to show that when SEF activity were higher on a trial, the animals perform better. 

Also, while greater SEF activity for higher incentive trials and for anti- vs pro-saccades are both consistent with the hypothesis that the SEF tracks cognitive effort, a more convincing test is whether the two dimensions interact in predicting SEF activity. An interaction is more convincing because, conceptually, these two dimensions are also correlated with arousal, independently, but high arousal is not the same thing as high effort. Effort, in contrast, is the conversion of motivation into action to overcome a challenge. Thus, as a result of effort, the effects of task difficulty will vary as a function of motivational state, and vice versa. IF SEF activity tracks effort, we should therefore expect that the effect of anti- vs pro-saccades on SEF activity should differ under high vs. low rewards (or, indeed that the effects of incentive magnitude on SEF activity should differ on anti- vs pro-saccade trials). 

Also, it is unclear whether effort follows the same low-frequency trends (downward, cross-session slopes) in pupil dilation and SEF activity. If pupil dilation and SEF activity are trending downwards across a session and they are also tracking effort, does that mean that monkeys are performing progressively worse across a session? If not, that seems to complicate the main inference. 

Minor comments

- In the results section, the Authors state that they "observed a slight disengagement of monkeys when the task is high-cognitive demanding … for a lower reward…" but they do not state their evidence of disengagement

- In Fig. 3, it is unclear why the shaded region between the anti- and pro-saccade conditions stops at 0,325 sec for the pupil diameter and at 4,0 sec for the CBV measurement?

- It wasn't clear why the Authors used a somewhat arbitrary threshold (the ratio of accuracy on anti- vs pro-saccades < 0.6) to retain trials for analyses. I would like to know the sensitivity of the results to this arbitrary threshold (would conclusions changed if they used values like 0.8 or 0.4, for example? It's also unclear why this threshold was used in the first place. Importantly, if the Authors only analyse sessions in which monkeys perform worse on anti-saccade trials by definition, then they are ensuring a positive correlation between errors and trial difficulty in a way that one would expect to covary with cognitive effort. That is, they are intentionally leaving out sessions where the presumably more difficult anti-saccade trials are performed equally well or better, and these sessions might provide and these sessions might be more revealing about if and how the SEF tracks cognitive effort. Finally, the Authors should stipulate the amount of data not retained for analyses according to their thresholding.

---

## [Decision Letter · Decision Letter 2]

15 Mar 2022

Dear Dr Pouget,

Thank you for submitting your revised Research Article entitled "Short and long time scales covariations between pupil diameter and Supplementary Eye Field activity" for publication in PLOS Biology. I have now obtained advice from the original reviewers and have discussed their comments with the Academic Editor. 

Based on the reviews (attached below), we will probably accept this manuscript for publication, provided you satisfactorily address the remaining points raised by Reviewers 3 and 4. Nevertheless, we do not think that further analyses are necessary and revisions to the text to address their concerns should be sufficient. Please also make sure to address the following data and other policy-related requests.

In addition, we would like to suggest a title that is more informative and appealing to a broad readership:

“Covariations between pupil diameter and Supplementary Eye Field activity suggest a role in cognitive effort implementation”

However, we would be happy to work with you on an alternative if you think our suggestion misrepresents your findings.

We expect to receive your revised manuscript within two weeks. 

*Published Peer Review History*

*Press*

Sincerely,

Ines

--

Ines Alvarez-Garcia, PhD

Senior Editor

PLOS Biology

ETHICS STATEMENT:

Thank you for including an ethics statement. Please include also the specific national or international regulations/guidelines to which your animal care and use protocol adhered. Note that institutional or accreditation organization guidelines (such as AAALAC) do not meet this requirement.

Fig 1B, C; Fig. 2A-K; Fig. 3; Fig. 4A-E; Fig. S1B and Fig. S2A-D.

BLURB

Please also provide a blurb which (if accepted) will be included in our weekly and monthly Electronic Table of Contents, sent out to readers of PLOS Biology, and may be used to promote your article in social media. The blurb should be about 30-40 words long and is subject to editorial changes. It should, without exaggeration, entice people to read your manuscript. It should not be redundant with the title and should not contain acronyms or abbreviations. For examples, view our author guidelines: https://journals.plos.org/plosbiology/s/revising-your-manuscript#loc-blurb

Reviewers' comments:

Rev. 1:

All my previous concerns have been adressed in great details by the authors, I am therefore happy to recommend the manuscript for publication. 

Rev. 2:

I command the authors for their revision. I now think that the manuscript quality is highly improved and ready for publication. I also think that the change in title and abstract enhances the impact of this work. Congratulations to the authors !

Rev. 3:

I have read the authors' response, and well as their revised manuscript. I appreciate the level of attention to my comments, and the efforts exerted by the authors to revise their manuscript. From my perspective, things are largely resolved, and I only have a few minor points. 

The abstract mentions the ACC without any context as to why this may be important. I'd suggest either deleting mention of the ACC in the abstract, or emphasizing how the SEF results contrast with negligible ACC signal changes. The ACC is also not defined in the abstract, nor upon first use in the introduction (line 55). 

Line 106 to 107, "...in order to ambition in future…" sounds a bit strange. Consider "… so that future.."

Lines 114 to 116 indicates that all trials were used in the analyses, regardless of whether they were correctly or incorrectly performed. This seems to contradict their responses to one of my concerns (Reviewer 3, 3rd main point in original review). I think all that is required is some qualifying statement in lines 114-116 (e.g., a statement to the effect that all data was used, unless noted otherwise). As an example lines 196-197 indeed do specify that only correct trials were used for a particular analysis. 

Line 154, "i.e." instead of "id est"?

I appreciate that the authors have rewritten much of the Discussion, and for the most part It reads well. The exception are the four paragraphs from lines 286-297, which read more like bullet points summarizing findings from four sets of studies. This portion of the Discussion could be improved with some edits to establish a cohesive narrative. 

Rev. 4:

In their revision, Claron and colleagues address some concerns and also provide additional analyses to address questions about whether the SEF and pupil dynamics track cognitive effort or not. On re-review, I agree with their decision to shift the emphasis (at least in the title) away from discussing cognitive effort. While there appears to be at least a nice long-term covariation between pupil dilation and SEF activity, and while changes in reward levels and demand level (anti- vs pro-saccade) seem to have similar impacts on pupil and SEF dynamics, evidence that these two features map to cognitive effort is relatively weak. Specifically, for example, there are no interactions between reward and demand level, as you might expect if these systems track the cognitive effort involved. There is no close, trial-by-trial coupling between task performance and pupil or SEF dynamics. The Authors do find that slow timescale performance drops off in the same way that pupil dilation and SEF response do, over trials, but this effect could also be explained by a general fatigue effect, or accommodation to the task over time that is not specific to changes in effort. In none of their analyses do the Authors show that a trial-level increase in SEF (because of, say, a temporary increase in endogenously or exogenously driven motivation levels) predicts trial-level changes in pupil dilation - much less that such changes predict better trial-level performance… 

A new concern has also arisen for me, in the Authors' response to a question about exclusion criteria, that also impacts on whether we can infer anything about effort from this dataset. Previously, the Authors had excluded data where ratio of accuracy on anti-vs pro-saccades was less than 0.6. When asked to consider all data, their response revealed a striking difference in the pattern of pupillary and SEF dynamics on anti- vs pro-saccades between the two monkeys in this study. Namely, while monkey S shows the expected pattern (pupil dilates more and performance is worse, on average, for anti-saccade trials, for most sessions), monkey G shows a very different pattern (pupil mostly dilates more, on average for pro-saccade trials, for most sessions, and average cross-session performance ratio is closer to 1.0 - indicating more similar performance between anti- and pro-saccade trials). In fact, excluding what look to be two outlier sessions, in terms of the ratio of dilation to anti- vs pro-saccade trials, monkey G appears to have, on average, a higher dilation to pro-saccade trials. What do the Authors make of the fact that, more often than not, monkey G's pupil dilates more for pro-saccade trials? In any case, monkey G's dilation patterns suggest that their dilation doesn't faithfully track effort because pro-saccade trials aren't harder that anti-saccade trials. On a related note, I would like to know what the Authors make of the huge difference in the ratio of pupil dilation on anti- vs pro-saccade trials for monkey G on the two sessions. What was so different about these sessions that could explain why the dilation ratio was so dramatically different than it was for all other sessions for monkey G?

This divergence between monkeys raises serious questions about what can be inferred about effort, because it is unclear - for monkey G at least - whether greater dilation reflects greater effort.

The complications with monkey G's data make me wonder how results would differ if the Authors analyzed monkey S's data only and asked whether trial-by-trial pupil dilation and SEF response to demand level or reward level predict better performance. If there is no strong coupling between trial-level data and performance, I recommend removing inferences about cognitive effort from the abstract. It is okay to speculate about it in the discussion, but it seems unwarranted to make claims about effort in the abstract when it is unclear how well the current experimental design is reliably sensitive to differences in effort. In fact, it seems that the design neither reveals reliable patterns of dilation / SEF activity across monkeys nor reliable patterns about effort from trial-to-trial. 

Instead, I think the abstract, and conclusions should be focused on the much more compelling inference that long-term changes in pupil dilation map to long-term changes in SEF activity, and that two seem more strongly coupled than do pupil dilation and ACC on this timescale.

---

## [Editor Report · Decision Letter 3]

4 Apr 2022

Dear Dr Pouget,

Thank you for submitting your revised Research Article entitled "Covariations between pupil diameter and Supplementary Eye Field activity suggest a role in cognitive effort implementation" for publication in PLOS Biology.

I have checked the revision and there are still several data and other policy-related requests that remain to be addressed. They are all included below my signature, so please read them carefully and submit all the information required.

We expect to receive your revised manuscript within one week. 

Sincerely,

Ines

Ines Alvarez-Garcia, PhD,

Senior Editor,

ialvarez-garcia@plos.org,

PLOS Biology

ETHICS STATEMENT:

Thank you for including an ethics statement. Please include also the specific national or international regulations/guidelines to which your animal care and use protocol adhered. Note that institutional or accreditation organization guidelines (such as AAALAC) do not meet this requirement. For example, some of these studies adhere to the recommendations of the Weatherall report.

Fig 1B, C; Fig. 2A-K; Fig. 3; Fig. 4A-E; Fig. S1B and Fig. S2A-D.

***In addition, please note that ALL data should be included in the manuscript or deposited in a publicly available database. Thus the data from the "SEF tracks cognitive effort" should be also made available before the manuscript can enter Production.

BLURB

Please also provide a blurb which (if accepted) will be included in our weekly and monthly Electronic Table of Contents, sent out to readers of PLOS Biology, and may be used to promote your article in social media. The blurb should be about 30-40 words long and is subject to editorial changes. It should, without exaggeration, entice people to read your manuscript. It should not be redundant with the title and should not contain acronyms or abbreviations. For examples, view our author guidelines: https://journals.plos.org/plosbiology/s/revising-your-manuscript#loc-blurb

---

## [Editor Report · Decision Letter 4]

29 Apr 2022

Dear Dr Pouget,

On behalf of my colleagues and the Academic Editor, Matthew Apps, I am happy to say that we can in principle accept your Research Article entitled "Covariations between pupil diameter and Supplementary Eye Field activity suggest a role in cognitive effort implementation" for publication in PLOS Biology, provided you address any remaining formatting and reporting issues. These will be detailed in an email that will follow this letter and that you will usually receive within 2-3 business days, during which time no action is required from you. Please note that we will not be able to formally accept your manuscript and schedule it for publication until you have completed any requested changes.

PRESS

Many congratulations and thank you again for choosing PLOS Biology for publication and supporting Open Access publishing. We look forward to publishing your study. 

Sincerely, 

Ines

--

Ines Alvarez-Garcia, PhD 

Senior Editor 

PLOS Biology
